# Admission High-Sensitive Cardiac Troponin T Level Increase Is Independently Associated with Higher Mortality in Critically Ill Patients with COVID-19: A Multicenter Study

**DOI:** 10.3390/jcm10081656

**Published:** 2021-04-13

**Authors:** Romaric Larcher, Noemie Besnard, Aziz Akouz, Emmanuelle Rabier, Lauranne Teule, Thomas Vandercamere, Samuel Zozor, Matthieu Amalric, Racim Benomar, Vincent Brunot, Philippe Corne, Olivier Barbot, Anne-Marie Dupuy, Jean-Paul Cristol, Kada Klouche

**Affiliations:** 1Biochemistry and Hormonology Department, Lapeyronie Hospital, University Hospital of Montpellier, 34090 Montpellier, France; s-zozor@chu-montpellier.fr (S.Z.); am-dupuy@chu-montpellier.fr (A.-M.D.); jp-cristol@chu-montpellier.fr (J.-P.C.); 2PhyMedExp, University of Montpellier, INSERM, CNRS, Arnaud de Villeneuve Hospital, University Hospital of Montpellier, 34090 Montpellier, France; k-klouche@chu-montpellier.fr; 3Intensive Care Medicine Department, Lapeyronie Hospital, University Hospital of Montpellier, 34090 Montpellier, France; n-besnard@chu-montpellier.fr (N.B.); amalric.matthieu@gmail.com (M.A.); r-benomar@chu-montpellier.fr (R.B.); v-brunot@chu-montpellier.fr (V.B.); p-corne@chu-montpellier.fr (P.C.); 4Intensive Care Unit, Hospital of Perpignan, 66000 Perpignan, France; aziz.akouz@ch-perpignan.fr (A.A.); lauranne.teule@ch-perpignan.fr (L.T.); barbot.olivier@ch-perpignan.fr (O.B.); 5Intensive Care Unit, Hospital of Narbonne, 11100 Narbonne, France; manuerabier@hotmail.fr (E.R.); thomasvandercamere@yahoo.fr (T.V.)

**Keywords:** COVID-19, SARS-CoV-2, high-sensitive cardiac troponin T, myocardial injury, outcomes, ICU

## Abstract

Background: In coronavirus disease 2019 (COVID-19) patients, increases in high-sensitive cardiac troponin T (hs-cTnT) have been reported to be associated with worse outcomes. In the critically ill, the prognostic value of hs-cTnT, however, remains to be assessed given that most previous studies have involved a case mix of non- and severely ill COVID-19 patients. Methods: We conducted, from March to May 2020, in three French intensive care units (ICUs), a multicenter retrospective cohort study to assess in-hospital mortality predictability of hs-cTnT levels in COVID-19 patients. Results: 111 laboratory-confirmed COVID-19 patients (68% of male, median age 67 (58–75) years old) were included. At ICU admission, the median Charlson Index, Simplified Acute Physiology Score II, and PaO_2_/FiO_2_ were at 3 (2–5), 37 (27–48), and 140 (98–154), respectively, and the median hs-cTnT serum levels were at 16.0 (10.1–31.9) ng/L. Seventy-five patients (68%) were mechanically ventilated, 41 (37%) were treated with norepinephrine, and 17 (15%) underwent renal replacement therapy. In-hospital mortality was 29% (32/111) and was independently associated with lower PaO_2_/FiO_2_ and higher hs-cTnT serum levels. Conclusions: At ICU admission, besides PaO_2_/FiO_2_, hs-cTnT levels may allow early risk stratification and triage in critically ill COVID-19 patients.

## 1. Introduction

Coronavirus disease 2019 (COVID-19), caused by severe acute respiratory syndrome coronavirus 2 (SARS-CoV-2) infection, has spread worldwide, resulting in a dramatic pandemic [1]. The number of COVID-19 cases is still dramatically increasing all around the world, and health systems confronting the pandemic have to face intensive care unit (ICU) resource scarcity [2]. Early risk stratification and identification of predictive factors of mortality in critically ill patients are therefore an important matter of concern in order to determine suitability for transfer to the ICU and to perform optimal follow-up and management.

In ICU settings, COVID-19 has initially been mainly associated with acute respiratory distress syndrome (ARDS), but evidence of multiple organ dysfunction, including myocardial injury, has been reported [3,4,5,6]. Cardiac disease in patients with COVID-19 has therefore been recognized as a common condition associated with a more severe clinical course and increased mortality [7,8]. In the same line, increases in cardiac troponin (cTn), also known as myocardial injury, have been reported to be associated with worse outcomes in patients with COVID-19 [8,9,10,11,12,13,14,15]. However, the prognostic value of cardiac biomarkers in COVID-19 critically ill patients have not yet been clearly defined. Obviously, data in the ICU population remains scarce, since almost all studies have involved a case mix of non-severe and severe patients, or few ICU patients [16,17,18,19].

The aim of this multicentric study was to assess the prognostic value of high-sensitive cardiac troponin T (hs-cTnT) levels at ICU admission in patients with critical SARS-CoV-2 infection.

## 2. Materials and Methods

### 2.1. Study Design and Settings

This observational retrospective cohort study was carried out from 9 March 2020 to 3 May 2020 in three French ICUs. Investigators on each site collected prospectively clinical and biological data for all critically ill patients diagnosed with COVID-19. Patients were followed up until hospital discharge or death and three months later by phone call.

### 2.2. Participants

All consecutive patients with a SARS-CoV-2 infection confirmed by RT-PCR who were at least 18 years old and admitted to the ICU during the study period were included. Patients with suspected COVID-19 but not confirmed by SARS-CoV-2 RT-PCR and those who did not have an hs-cTnT measurement within the first 24 h of ICU admission were excluded.

### 2.3. Data Collection

Demographical and clinical data, including morbidities, were collected. Patients’ prior health status was evaluated by the Charlson Comorbidity Index [20]. Chronic heart disease was defined as a medical history of myocardial infarction, coronary ischemia treated by angioplasty and/or stents, cardiac surgery, or chronic heart failure (i.e., exertional or paroxysmal nocturnal dyspnea that has responded to digitalis, diuretics, or afterload-reducing agents). The severity of disease was assessed 24 h after ICU admission using the Simplified Acute Physiology Score (SAPS) II [21]. At ICU admission, the oxygen arterial partial pressure to the fraction of inspired oxygen (PaO_2_/FiO_2_) ratio [22] was recorded. Biological data were also collected at admission, particularly hs-cTnT, C reactive protein (CRP), and D-dimer. The D-dimer level was determined using D-Dimer HS 500 assay on ACL TOP 700 analyzer (Werfen, Barcelona, Spain). Procalcitonin (PCT) was determined using the BRAHMS PCT assay based on TRACE (Time-Resolved Amplified Cryptate Emission) on the Kryptor Gold^©^ instrument (ThermoScientific, BRAHMS AG, Hennigsdorf, Germany). All other biological parameters, such as hs-cTnT, serum creatinine, CRP, and lactate, were performed on a Cobas 8000 instrument using e802 and c702 modules (Roche Diagnostic, Meylan, France).

Within the first five days of ICU admission, a transthoracic echocardiography was performed using a Vivid S70 ultrasound machine (GE Healthcare, Horton, Norway), equipped with a M5S (1.4–4.6MHz) cardiac probe allowing M-mode and two-dimensional measurements. Ventricular systolic function was assessed by measuring the ejection fraction (EF) from the mean of three measurements in different cardiac cycles using the biplane disc summation method and by calculation of fractional shortening, from the reduction of the right ventricular (LV) internal diameter during the cardiac cycle (modified Simpson method). Tricuspid annular plane systolic excursion (TAPSE) measurements were also performed using M-mode. Left ventricular dysfunction was defined as a left ventricular ejection fraction (LVEF) < 50%, whereas right ventricular dysfunction was defined with a TAPSE measurement < 15 mm. A cardiac dysfunction was considered to be present in case of left or/and right ventricular dysfunction.

Therapies instituted during ICU stay, including vasoactive drugs, invasive mechanical ventilation, renal replacement therapy, and extracorporeal membrane oxygenation (ECMO), were recorded.

The ICU length of stay and the outcome, including ICU, in-hospital, and 90-day mortalities, were recorded. Predictive factors of in-hospital mortality were then identified.

### 2.4. Statistical Analysis

Data are described as median and interquartile range (IQR) or number and percentage. The population was divided into two groups according to vital status at hospital discharge. Categorical variables were compared using the Chi-square test and continuous variables using the nonparametric Wilcoxon test. Mortality is displayed as Kaplan–Meier plots and compared using log-rank tests. Risk factors associated with in-hospital mortality and need for invasive mechanical ventilation were assessed using univariable and multivariable cox regression. For the multivariable analysis, variables were selected according to their statistical significance in univariable analysis (with a *p*-value < 0.2). Because of skewed distributions, biomarker values were log-transformed before modeling. Proportional hazard assumption was assessed by inspecting the scaled Shoenfeld residuals and regression splines. Hazard ratios (HRs) were given with a 95% confidence interval. The sensitivity and specificity of the factors associated with in-hospital mortality were assessed and receiver operating characteristic (ROC) curves were plotted. The optimal cut-off values were estimated by maximizing the Youden index. The risk factors associated with the occurrence of cardiac dysfunction were assessed using a logistic regression model. All tests were two-sided and a *p*-value less than 0.05 was considered statistically significant. R software version 4.0.2 (Free Software Foundation, Boston, MA, USA) was used for analyses.

## 3. Results

### 3.1. Study Population

Among the 184 patients admitted to ICUs with suspected COVID-19, 73 were excluded: 14 because of non-confirmed diagnosis and 59 because of missing data (Figure 1). One hundred and eleven patients (75 males, 68%) with a median age of 67 (58–75) years were then included in the study. As shown in Table 1, the median Charlson Index was 3 (2–5) and most of the patients (48%) had at least two comorbid conditions such as hypertension (47%), diabetes (31%), chronic heart diseases (20%), chronic pulmonary disease (20%), or chronic kidney disease (13%).

At ICU admission, the median SAPS II was 37 (27–48). The severity of patients’ illnesses was mainly related to respiratory failure, given that the median PaO_2_/FiO_2_ ratio at admission was 140 (98–154) and severe acute respiratory distress syndrome (ARDS) according to the Berlin definition occurred in almost one third of the patients (31%). Almost all patients (83%) included in the study were admitted to the ICU from the emergency department or after a hospital stay in the ward of ≤ 24 h.

The biological data at admission are summarized in Table 1. A huge elevation of CRP and D-dimer was observed at 153 (112–222) mg/L and 890 (572–1950) ng/mL, respectively. Levels of cTnT were increased over the 99th percentile (>14 ng/L) in 61 patients (55%) and above 52 ng/L in 18 patients (16%). The PCT values were low and mostly under the cutoff value of 0.5 ng/mL (median PCT at 0.4 (0.2–0.9) ng/mL).

### 3.2. Outcomes

During ICU stay, more than two third of patients (75 patients, 68%) required invasive mechanical ventilation, and among them, 38 (34% of the whole population and 51% of mechanically ventilated patients) needed—in addition—a prone positioning, and three required a veno-venous ECMO (3%) for refractory hypoxemia. Acute kidney injury at a KDIGO stage ≥ 1 occurred in 66 patients (60%) and 17 (15%) underwent renal replacement therapy. Norepinephrine (≥1 µg/kg/min) was administered in 41 (37%) patients, mainly in the case of vasoplegic shock, except in five patients who had mixed cardiogenic and vasoplegic shock. Dobutamine was administered in one patient. A bacterial infection (six pneumonia, two bacteremia, and one urosepsis) was documented concomitantly with the SARS-CoV-2 infection in nine patients treated with norepinephrine.

In 66 patients, echocardiography was performed, and the median LEVF was 60% (50–60%). A cardiac dysfunction was diagnosed in 15 (23%) of these 66 patients (Appendix A). Two patients had pulmonary embolism without right ventricular dysfunction, shock, or hs-cTnT elevation.

Thirty-two patients died before hospital discharge, showing a mortality rate of 29%. Median ICU and hospital lengths of stay were five (2–13) and 27 (14–49) days; three months later, all discharged patients from the hospital were still alive (Table 1). The ninety-day survival probability was 72% (64–81%).

### 3.3. Factors Associated with In-Hospital Mortality

By univariable analysis, age, sex, and a high Charlson Index were significantly associated with in-hospital mortality. At admission, higher SAPS II, a lower PaO_2_/FiO_2_ ratio, and increased serum creatinine and hs-cTnT levels were also significantly associated with mortality (Table 1). During ICU stay, the requirement for vasoactive drugs and/or renal replacement therapy significantly worsen the outcome.

By multivariable analysis, only a low PaO_2_/FiO_2_ ratio and an elevated hs-cTnT serum level among the variables studied were independently associated with in-hospital mortality (Table 2).

Of note, the analysis of the 66 patients in which echocardiography was performed failed to demonstrate any association between hs-cTnT serum levels and cardiac function (*p* = 0.12) (Appendix A).

### 3.4. ROC Curves of the Main Prognostic Factors of Mortality at ICU Admission

At ICU admission, age, hs-cTnT, and the PaO_2_/FiO_2_ ratio area under the curves (AUCs) for predicting in-hospital mortality were 0.794 (95% CI, 0.705–0.883), 0.792 (95% CI, 0.702–0.882), and 0.696 (95% CI, 0.589–0.802), respectively. The D-dimer and CRP levels showed an ROC-AUC of 0.609, (95% CI, 0.485–0.734) and 0.539 (95% CI, 0.410–0.669), respectively. The ROC curves are displayed in Figure 2.

Hs-cTnT has a sensitivity and a specificity at 69% and 79%, respectively, to predict in-hospital mortality using a cutoff value at 22 ng/L (Table 3). Age and the PaO_2_/FiO_2_ ratio had a higher specificity (81% and 82%, respectively) but lower sensitivity (63% and 50%, respectively) with a cutoff value at 74 years old and 100, respectively. D-dimer, at a cutoff value of 500 ng/L, showed the highest sensitivity (96%) but the lowest specificity (26%), and D-dimer ≥ 500 ng/L also had the highest negative predictive value (NPV) at 94%, whereas hs-cTnT ≥ 22 ng/L, age ≥ 74 years old, and PaO_2_/FiO_2_ ratio ≤ 100 had an NPV of 86%, 84%, and 80%, respectively. All factors had low positive predictive values (PPVs).

Ninety-day survival was significantly altered according to the level of hs-cTnT and the PaO_2_/FiO_2_ ratio. The corresponding Kaplan–Meier curves are displayed in Figure 3.

### 3.5. Factors Associated with Invasive Mechanical Ventilation Needs

By univariable analysis, age, high Charlson Index and SAPS II, a low PaO_2_/FiO_2_ ratio, need for vasoactive drugs, and increased serum hs-cTnT levels were significantly associated with invasive mechanical ventilation (Appendix A).

By multivariable analysis, only higher SAPS II was independently associated with invasive mechanical ventilation (Appendix A). None of studied biological variables were independently associated with the need for invasive mechanical ventilation.

## 4. Discussion

In this multicentric study, including 111 critically ill laboratory-confirmed COVID-19 patients, we observed an in-hospital mortality rate of 29%. We found that a low PaO_2_/FiO_2_ ratio and elevated hs-cTnT serum levels at ICU admission were independently associated with in-hospital mortality. All variables had a fair negative but low positive value to predict in-hospital mortality.

Previous studies have already reported that a lower PaO_2_/FiO_2_ ratio is associated with mortality in critical COVID-19 patients [1,12,23,24]. A lower PaO_2_/FiO_2_ ratio is also recognized as the major determinant predicting morbidity and mortality in patients with non-COVID-19 ARDS [22,25,26]. In addition, we showed in this study that an increase in hs-cTnT is associated with in-hospital mortality in critically ill COVID-19 patients. There is evidence of an association between cTnI levels and mortality in mild–severe COVID-19 patients [9,16,19,27]. Moreover, recent meta-analyses have concluded that increased cTnI levels are significantly associated with the most severe forms of COVID-19 [14,28]. Our observations confirmed these previous reports and are in line with the results recently reported by Demir et al. [29]. However, conflicting results have been reported by Metkus et al. [30], highlighting that the association between cTn elevation and in-hospital mortality of mechanically ventilated COVID-19 patients was weak after adjustment for cofounders, namely, age, sex, and multi-system organ dysfunction. Though they take into account normal values accordingly, the use of both cTnT and cTnI measurements may have introduced some bias in their results [30]. Nonetheless, in our study, hs-cTnT levels, as well as PaO_2_/FiO_2_, were independently associated with mortality, given that other organ dysfunctions, age, and gender were included in the multivariable analysis.

In critically ill patients with SARS-CoV-2 infection, several mechanisms could be involved in myocardial injury and the subsequent elevation of hs-cTn levels [31]. Acute myocardial infarction (types 1 and 2) may complicate COVID-19, as reported in acute bacterial or viral infections [32]. In our cohort, though hs-cTnT levels were frequently over the 99th percentile (>14 ng/L), they rarely crossed 52 ng/L, the cut-off value for myocardial infarction diagnosis [33]. In addition, electro/echocardiographic and hs-cTnT kinetics along ICU stay ruled out myocardial necrosis. COVID-19-associated myocarditis may also induce an increase in troponin [8,31,34,35,36]. The clinical features of acute myocarditis are heterogeneous, from new-onset heart failure to arrhythmic events, or infarct-like symptoms with usually preserved LVEF [37]. In our patients, the occurrence of acute myocarditis with preserved systolic function could explain the increase in hs-cTnT without echocardiographic abnormalities. Actually, there are limited reports on SARS-CoV-2-associated myocarditis, and a few autopsy series have shown virus in the myocardium [34,38]. Nonetheless, sepsis, cardiac adrenergic hyperstimulation, and pulmonary embolism could enhance troponin liberation [30,31,35]. Moreover, comorbid conditions such as coronary artery disease, chronic ischemic cardiomyopathy, or hypertension, which are common in COVID-19 patients [7,12,13], may also contribute to troponin elevation. Regardless, COVID-19 led to inflammation, endothelial dysfunction, hypercoagulation state, hypoxia, and hemodynamic instability that may induce myocardial injury and, consequently, troponin elevation [8,10,13,16,34,38,39].

Our results are in line with those previously reported in septic critically ill patients regarding both cTnT and cTnI levels [40,41,42,43], suggesting that cTnT levels may add to risk stratification and in a better manner than inflammatory or thrombotic markers in COVID-19 patients [9,16,17,18,19]. Though a recent meta-analysis have highlighted the prognostic value of increased CRP and D-dimer plasma levels in non-severe and severe COVID-19 [44], the value of such markers in ICU settings is flawed, given that they are frequently increased, as we observed in our patients [16,45]. Our results should prompt intensivists to pay attention to hs-cTnT levels, in addition to the PaO_2_/FiO_2_ ratio, among critically ill patients admitted with SARS-CoV-2 infection for triage and risk stratification. A higher admission severity score may also help to predict the need for invasive mechanical ventilation.

We must acknowledge some limitations to this study. First, it was limited by its retrospective design, inducing bias in data collection and result interpretation. Some hs-cTnT missing values forced us to exclude 59 patients, but this work is the largest multicenter analysis on the prognostic value of cTn in ICU settings. Second, hs-cTnT levels were not systematically monitored during ICU stay. Obviously, hs-cTnT kinetics would be very informative in both diagnosis and prognosis in this population. Last, echocardiography was performed in a small subset of included patients. Moreover, endomyocardial biopsy and cardiac magnetic resonance imaging might have been useful for diagnosis of acute myocarditis, particularly for those with preserved systolic function.

## 5. Conclusions

In conclusion, elevated troponin plasmatic concentrations are commonly observed in critically ill COVID-19 patients. We showed that, at admission, myocardial injury and a low PaO_2_/FiO_2_ ratio were strongly associated with in-hospital mortality. These variables may help for an early and best risk stratification in such patients. Further studies are, however, mandatory to confirm our results.

## Figures and Tables

**Figure 1 jcm-10-01656-f001:**
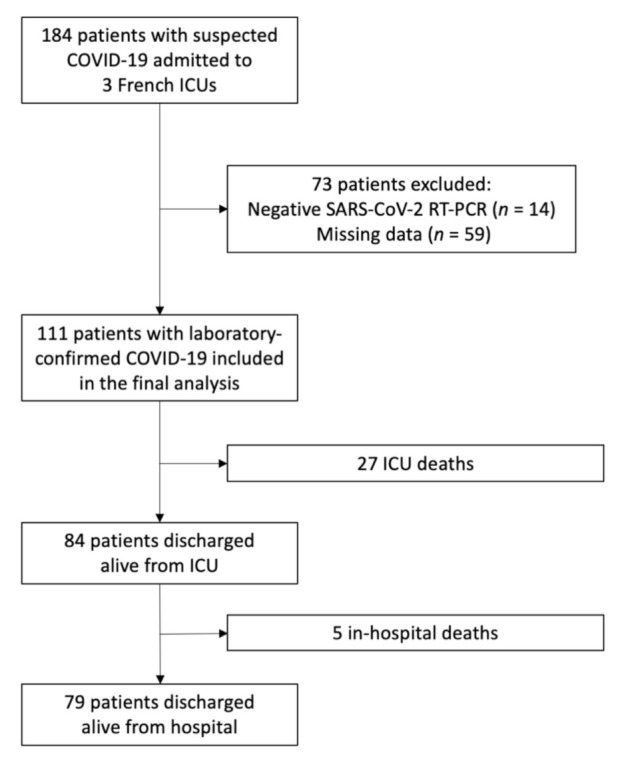
Flow chart of the study population. ICU, intensive care unit; SARS-CoV-2, severe acute respiratory syndrome coronavirus 2.

**Figure 2 jcm-10-01656-f002:**
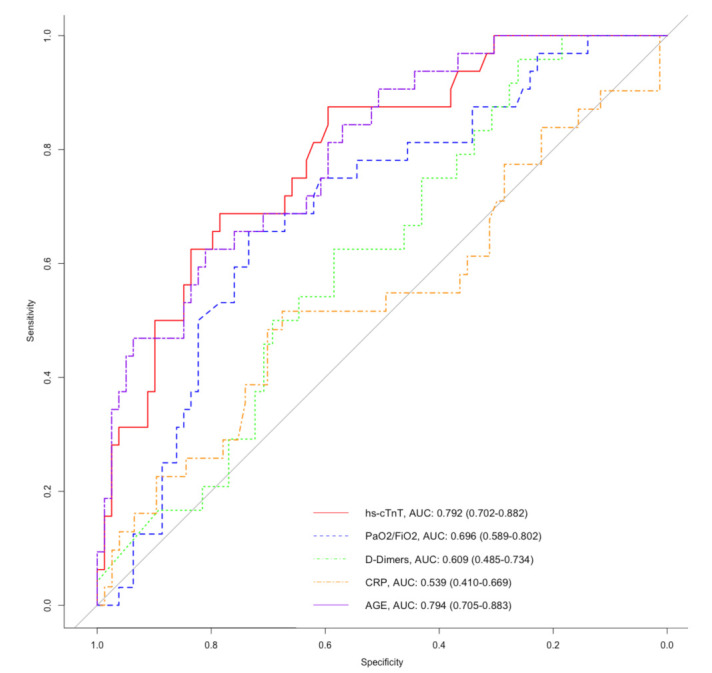
Receiver operating characteristic (ROC) curves of age (purple semi-dashed line), initial PaO_2_/FiO_2_ ratio (blue dashed line), initial high-sensitive cardiac troponin T (hs-cTnT, red line), initial D-Dimer (green dotted line), and initial C reactive protein (CRP, orange dashed and dotted line) for predicting in-hospital mortality in critically ill patients with COVID-19.

**Figure 3 jcm-10-01656-f003:**
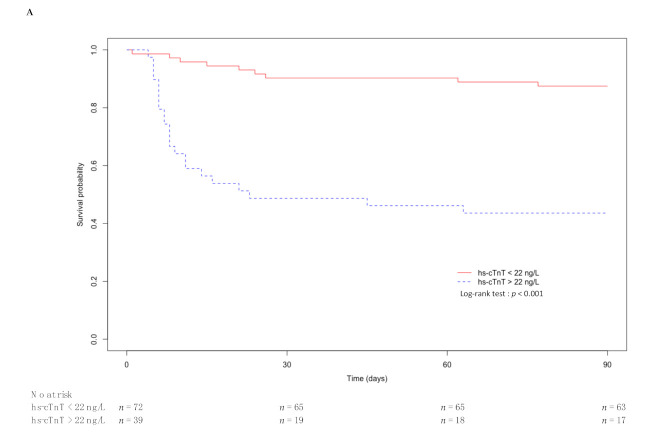
Kaplan–Meier curve of 90-day survival after ICU admission of critically ill patients with COVID-19: (**A**) hs-cTnT < 22 ng/L versus hs-cTnT > 22 ng/L; (**B**) no acute respiratory distress syndrome (ARDS) (PaO_2_/FiO_2_ >300) versus mild (PaO_2_/FiO_2_ = 201–300), moderate (PaO_2_/FiO_2_ = 101–200), and severe ARDS (PaO_2_/FiO_2_ ≤ 100).

**Table 1 jcm-10-01656-t001:** Baseline characteristics of the study population and mortality.

	Total (*n* = 111)	Non-Survivor (*n* = 32)	Survivor (*n* = 79)	*p*-Value
Age (years), median (IQR)	67 (58–75)	75 (67–81)	63.0 (56–72)	<0.001
Male, *n* (%)	75 (68%)	16 (50%)	59 (75%)	0.01
BMI ^1^ (kg/m²), median (IQR)	28 (25–32)	26 (23–31)	28 (25–32)	0.57
Hypertension, *n* (%)	52 (47%)	18 (56%)	34 (43%)	0.21
Diabetes, *n* (%)	33 (31%)	12 (38%)	22 (28%)	0.32
Chronic heart disease, *n* (%)	22 (20%)	10 (31%)	12 (15%)	0.06
Chronic kidney disease, *n* (%)	14 (13%)	8 (25%)	6 (8%)	0.02
Chronic pulmonary disease, *n* (%)	22 (20%)	11 (34%)	11 (14%)	0.02
Charlson Index, median (IQR)	3 (2–5)	5 (4–8)	3 (2–4)	<0.001
SAPS II ^2^, median (IQR)	37 (27–48)	43 (34–54)	33 (25–46)	0.01
PaO_2_/FiO_2_, median (IQR)	140 (98–154)	100 (85–139)	160 (105–206)	0.003
PaO_2_/FiO_2_ ≤100, *n* (%)	34 (31%)	17 (53%)	17 (22%)	0.002
Invasive mechanical ventilation, *n* (%)	75 (68%)	25 (78%)	50 (63%)	0.14
Duration (days), median (IQR)	10 (5–19)	9 (6–16)	12 (5–18)	-
Prone positioning, *n* (%)	38 (34%)	14 (44%)	24 (30%)	0.02
ECMO ^3^, *n* (%)	3 (3%)	1 (3%)	2 (3%)	0.86
Duration (days), median (IQR)	17 (14–20)	-	-	-
Norepinephrine (>1 µg/kg/min), *n* (%)	41 (37%)	17 (53%)	24 (30%)	0.03
Duration (days), median (IQR)	7 (4–12)	9 (5–15)	5 (3–10)	-
RRT ^4^, *n* (%)	17 (15%)	10 (31%)	7 (9%)	0.005
Duration (days), median (IQR)	14 (4–31)	10 (3–19)	22 (10–38)	-
Creatininemia (µmol/L), median (IQR)	84 (62–111)	92 (72–168)	82 (57–101)	0.01
hs-cTnT ^5^ (ng/L), median (IQR)	16.0 (10.2–31.9)	35.0 (16.8–106.0)	12.5 (7.5–20.0)	<0.001
CRP ^6^ (mg/L), median (IQR)	153 (112–222)	134 (93–215)	154 (115–223)	0.45
PCT ^7^ (ng/mL), median (IQR)	0.4 (0.2–0.9)	0.6 (0.3–1.3)	0.3 (0.2–0.9)	0.11
Lactatemia (mmol/L), median (IQR)	1.3 (0.9–1.7)	1.3 (0.8–1.8)	1.3 (1.0–1.6)	0.62
D-dimer (ng/mL), median (IQR)	890 (572–1950)	1340 (658–1958)	859 (497–1861)	0.09
LVEF ^8^ (%), median (IQR)	60 (50–60)	55 (50–60)	60 (50–60)	0.28
TAPSE ^9^ (mm), median (IQR)	18 (16–22)	18 (14–23)	18 (17–22)	0.49
ICU LOS ^10^ (days), median (IQR)	10 (6–18)	8 (5–17)	10 (6–18)	-
Hospital LOS (days), median (IQR)	19 (11–31)	10 (7–22)	21 (13–34)	-
Death, *n* (%)	32 (29%)	-	-	-

^1^ BMI, body mass index; ^2^ SAPS II, Simplified Acute Physiology Score II; ^3^ ECMO, extracorporeal membrane oxygenation; ^4^ RRT, renal replacement therapy; ^5^ hs-cTnT, high-sensitive cardiac troponin T; ^6^ CRP, C reactive protein; ^7^ PCT, procalcitonin; ^8^ LVEF, left ventricular ejection fraction (evaluated in a subset of 66 patients); ^9^ TAPSE, tricuspid annular plane systolic excursion (evaluated in a subset of 66 patients); ^10^ ICU LOS, intensive care unit length of stay.

**Table 2 jcm-10-01656-t002:** Factors associated with in-hospital mortality in critically ill COVID-19 patients.

	Univariable Analysis Hazard Ratio (IC 95%)	*p*-Value	Multivariable Analysis Hazard Ratio (IC 95%)	*p*-Value
Age	1.1 (1.1–1.2)	0.01	1.06 (0.99–1.14)	0.11
Male sex	0.4 (0.2–0.81)	<0.001	0.50 (0.21–1.05)	0.64
BMI ^1^	0.99 (0.93–1.1)	0.68		
Smokers	0.5 (0.068–3.6)	0.49		
Charlson Index	1.3 (1.2–1.4)	<0.001	4.46 (0.26–75.55)	0.30
SAPS II ^2^	10 (1.7–58)	0.01	1.06 (0.08–13.36)	0.96
PaO_2_/FiO_2_ ≤100	3.4 (1.7–6.8)	<0.001	4.65 (1.81–11.97)	0.001
Invasive mechanical ventilation	1.8 (0.76–4.1)	0.19	2.88 (0.76–10.97)	0.12
Norepinephrine (>1 µg/kg/min)	2.1 (1–4.1)	0.04	1.08 (0.30–2.84)	0.90
Renal replacement therapy	2.9 (1.4–6.2)	0.005	1.58 (0.58–4.31)	0.37
Creatininemia (log10)	8.5 (2.3–32)	0.002	1.45 (0.20–10.73)	0.71
Hs-cTnT ^3^ (log10)	4.3 (2.5–7.2)	<0.001	4.96 (1.92–12.86)	<0.001
CRP ^4^ (log10)	0.83 (0.37–1.9)	0.65		
PCT ^5^ (log10)	1.7 (1–2.9)	0.05	1.60 (0.60–4.25)	0.3
Lactatemia (log10)	1.8 (0.21–16)	0.58		
D-dimer (log10)	2.4 (0.81–6.9)	0.12	1.21 (0.17–3.95)	0.8

^1^ BMI, body mass index; ^2^ SAPS II, Simplified Acute Physiology Score II; ^3^ hs-cTnT, high-sensitive cardiac troponin T; ^4^ CRP, C reactive protein; ^5^ PCT, procalcitonin.

**Table 3 jcm-10-01656-t003:** Sensibility, specificity, positive, and negative predictive values for mortality prediction in critically ill patients with COVID-19.

	Sensibility (%)	Specificity (%)	PPV ^1^ (%)	NPV ^2^ (%)
age ≥ 74 years old	63	81	57	84
hs-cTnT ^3^ ≥ 22 ng/L	69	79	56	86
PaO_2_/FiO_2_ ≤ 115	66	73	50	84
PaO_2_/FiO_2_ ≤ 100	50	82	53	80
D-dimer ≥ 500 ng/L	96	26	34	94
CRP ^4^ ≥ 135 mg/L	52	68	39	78

^1^ PPV, positive predictive value; ^2^ NPV, negative predictive value; ^3^ hs-cTnT, high-sensitive cardiac troponin T; ^4^ CRP, C reactive protein. The optimal cutoff value of each variable estimated by maximizing the Youden index was selected, except for PaO_2_/FiO_2_ ≤ 100, which is the threshold of the Berlin definition for severe acute respiratory distress syndrome.

## Data Availability

The authors consent to share the collected data with others. Data will be provided to qualified investigators free of charge, after careful examination of required documents (summary of the research plan, request form, and IRB approval) by the study board of investigators. Data will be available immediately after the main publication and indefinitely.

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
