# Peer review of "Admission High-Sensitive Cardiac Troponin T Level Increase Is Independently Associated with Higher Mortality in Critically Ill Patients with COVID-19: A Multicenter Study"

_jcm, 2021, doi:10.3390/jcm10081656_

Round 1

Reviewer 1 Report

The present manuscript includes findings from >100 COVID-19 pneumonia patients hospitalized at 3 ICU centers. Marker of myocardial injury troponin T was measured at admission, and was found to be independently associated with in-hospital mortality. The study is overall well written and investigates and interesting topic. The statistical analysis would require some adjustments. I have the following comments:

  • Several studies addressed this same topic with comparable results (i.e. Ramazan Gunduz et al. journal thrombosis and throm 2021; Ozan M. Demir et al. Am j Card 2021). To improve originality, the authors could provide further analysis, in example by including an investigation of mechanical ventilation predictors
  • Statistical analysis:
    1. biomarkers values (troponin, CRP, D-Dimer etc.) likely have a skewed distribution, and should be changed into corresponding Log-transformed when performing regression analysis
    2. Authors should include in table univariable analysis results for those variables that where analyzed. Furthermore, by step-wise regression method, a final model including only significant variables should be reported (i.e. male sex appears to be non significant)
    3. The term multivariate is used throughout the manuscript. A multivariate analysis is actually when more than one dependent variable is analyzed. The term is often (and erroneously) used in medical papers but I suggest the authors to change multivariate to the (correct) term multivariable
    4. Authors should be complimented for having carefully assessed in-hospital mortality timing. However, in these cases, Cox regression analysis (with hazard ratios calculation) rather than logistic regression analysis might be a better option to assess predictors of mortality

Reviewer 2 Report

High sensitivity Troponin can be released 

Round 2

Reviewer 1 Report

Thank you for properly addressing all the points. I just have one minor suggestion regarding presentation: I would specify, within Cox-regression tables, when a log-transformed variable was used (i.e. hsTropT (Log10)) in order to provide a better understanding of reported HR.

Reviewer 2 Report

Troponin  I and T levels are increased in conditions other than acute coronary syndromes including pulmonary embolism, sepsis, arrhythmias, Heart Failure, ecc.

The correlation with other markers of risk is the most interesting aspect of this paper.

Unfortunally 73 patients were excluded

hs-cTnT can increase in renal failure and 17 patients underwent renal replacement therapy. Can you provide data on these patients hs-cTnT?

Chronic heart disease is a general definition

hs-cTnT and D-dimer were not measured  systematically. Do you have other data un following samples of D-dimer?

Motivation for noradrenalin administration were not presented

There is any relation between D-dimer, hs-cTnT and with TAPSE or right ventricular function measured with other methods?

Pulmonary Blood pressure were measured or calculated from echo Doppler exams?

Any patient suffered from pulmonary embolism? If yes hs-cTnT levels were particularly elevated?

Do you can comment on differences between hs-cTnT and hs-cTnI in ICU setting and in Covid-19?
